# Codon usage bias controls mRNA and protein abundance in trypanosomatids

**Laura Jeacock, Joana Faria, David Horn\***

Wellcome Trust Centre for Anti-Infectives Research, School of Life Sciences, University of Dundee, Dundee, United Kingdom

**Abstract** Protein abundance differs from a few to millions of copies per cell. *Trypanosoma brucei* presents an excellent model for studies on codon bias and differential gene expression because transcription is broadly unregulated and uniform across the genome. *T. brucei* is also a major human and animal protozoal pathogen. Here, an experimental assessment, using synthetic reporter genes, revealed that GC3 codons have a major positive impact on both mRNA and protein abundance. Our estimates of relative expression, based on coding sequences alone (codon usage and sequence length), are within 2-fold of the observed values for the majority of measured cellular mRNAs (n > 7000) and proteins (n > 2000). Our estimates also correspond with expression measures from published transcriptome and proteome datasets from other trypanosomatids. We conclude that codon usage is a key factor affecting global relative mRNA and protein expression in trypanosomatids and that relative abundance can be effectively estimated using only protein coding sequences.
DOI: https://doi.org/10.7554/eLife.32496.001

**\*For correspondence:**
d.horn@dundee.ac.uk

**Competing interests:** The authors declare that no competing interests exist.

## Introduction

Cellular growth and function depend upon the efficient expression of a large number of proteins that differ in abundance over a range from only a few molecules to millions of molecules per cell. Gene expression can be controlled at many levels, including transcription, mRNA stability, translation and protein stability. *Trypanosoma brucei* are parasitic protozoa that present a unique model for studies on gene expression because a relatively small number of 'dispersed' GT-rich RNA polymerase II (pol-II) promoters drive constitutive polycistronic transcription (*Wedel et al., 2017*). Indeed, *T. brucei* have no known regulated RNA polymerase II promoters for protein-coding genes. Some stress response genes and cell cycle regulated genes are located towards the ends of the polycistronic units (*Kelly et al., 2012*) but there is otherwise limited clustering of functionally related genes. In addition, trypanosomes have only two genes containing known introns (*Mair et al., 2000*), and every mRNA has an identical sequence *trans*-spliced onto the 5'-end (*Clayton, 2016*). These features are conserved in the related parasitic trypanosomatids, *Trypanosoma cruzi* and *Leishmania spp.* and in other trypanosomatids. This remarkable level of uniformity in terms of transcription and mRNA processing indicates that gene expression control primarily operates post-transcription.

Sixty-one alternative base-triplets (codons) in DNA and mRNA encode for twenty different amino acids, such that many amino acids are encoded by two or up to six distinct but 'synonymous' codons. These codons can vary in their first position for three amino acids and in their third position for eighteen. Although recognised several decades ago, our understanding of the impact of inherently redundant codon usage and codon usage bias remains incomplete. One mRNA can yield 4000 molecules of protein, as measured in yeast (*Futcher et al., 1999*), while the average protein:mRNA ratio in insect-form *T. brucei* is estimated to be 550:1; median values of three and 1650 molecules per gene, per cell, respectively (*Kolev et al., 2010*). Thus, there is substantial capacity for expression control at the level of translation. Indeed, individual codons can control translation-rate in yeast

**eLife digest** Genes are made up of DNA, which contains all the information and instruction needed to build an organism. This information is stored as a genetic code consisting of four bases: adenine (A), cytosine (C), guanine (G) and thymine (T). The order of these bases and their different combinations serves as a blueprint for making thousands of different proteins and to assemble living cells.

Proteins are vital for almost every process that keeps cells alive. They are made up of chains of small molecules called amino acids, via a process that takes several steps. First, the code from the gene needs to be copied into so-called messenger RNA molecules (mRNA). Cells then decode these mRNAs by reading the bases in groups of three, called codons. Most codons specify an amino acid, though some mark the start and end point of the protein.

Cells need different proteins in varying amounts. In fact, a given protein may be present in only a few copies in one cell, while another may be present in millions of copies, and in both cases the amount of protein present often needs to be tightly controlled.

One way to control the number of copies of a protein that are made from a mRNA is via the specific choice of codons. Most amino acids are encoded by more than one codon and using a different codon can lead the mRNA to yield more or less protein without actually changing the protein that is made.

Researchers are particularly interested in protein production in parasites known as trypanosomes, which cause a range of diseases in humans and other animals. Although most aspects of protein production are similar in humans and in trypanosomes, the differences indicate that codon choice would be particularly important in the parasites. However, until now it was not known how codon choice contributes to the control of the copy number of proteins in these parasites.

To test this, Jeacock, Faria and Horn engineered fully synthetic genes for fast or slow protein production and inserted them into the parasite's genome. The genes generated the proteins at the predicted levels. An analysis of several datasets of protein abundance showed that the number of proteins in related parasites could also be predicted by using only the DNA code of the gene. Jeacock et al. could even predict the number of mRNAs in a parasite cell. A separate study by Nascimento, Kelly et al. also showed that codon identity predicts mRNA levels in trypanosomes.

These findings will help researchers to predict mRNA and protein levels in these parasites using simple computer models. A better understanding of gene expression in these parasites will also improve our knowledge of the fundamental aspects of protein production in relation to evolution, biotechnology and disease.

DOI: https://doi.org/10.7554/eLife.32496.002

(*Gardin et al., 2014*), codon-dependent local translation slowdown can facilitate nascent peptide processing in eukaryotes (*Mahlab and Linial, 2014*; *Pechmann et al., 2014*) and changes in relative codon usage and cognate tRNA abundance can control differentiation-related translation pro-grammes in metazoa (*Gingold et al., 2014*). Thus, different codons are decoded at different rates by native ribosomes (*Hanson and Coller, 2018*; *Novoa and Ribas de Pouplana, 2012*; *Quax et al., 2015*). Codon usage can also impact mRNA decay (*Hanson and Coller, 2018*; *Presnyak et al., 2015*).

In trypanosomatids, highly expressed genes are amplified in tandem and are enriched in GC3 codons, those codons that have a G or a C at the third position; cognate tRNA genes for these codons also display increased copy number (*Horn, 2008*). Although pol-II promoters appear to underpin much of the regulation of gene expression in metazoa and other eukaryotes, GC3 codons are also favoured in highly expressed genes in mammals and the evidence suggests that protein abundance is in fact predominantly controlled at the level of translation (*Schwanhäusser et al., 2011*). In trypanosomatids, control of translation and mRNA stability are typically ascribed to 3'-untranslated sequences and their interactions with RNA-binding proteins and indeed, such regula-tion does operate (*Clayton, 2013*), but informatics analysis also supports a role for translational selection, the increased translation of GC3 codons (*Chanda et al., 2007*; *Horn, 2008*; *Subramanian and Sarkar, 2015*).

Since transcription for all but a few genes is polycistronic and constitutive, trypanosomatids present excellent model eukaryotic systems in which to investigate the impact of codon bias on gene expression control. Here, we combine experimental and bioinformatics analyses to explore the contribution that codon usage makes to differential gene expression in *T. brucei* and in other trypanosomatids. Our results indicate a capacity for codon bias based control of relative protein abundance over several orders of magnitude. We find that mRNA abundance is also increased by favoured codons. Remarkably, our predictions, based on trypanosomatid protein-coding sequences alone, correlate well with observed measures of steady-state mRNA and protein abundance.

## Results

### GC3 codon bias increases protein expression

Prior analysis of codon usage in trypanosomatids indicated a correlation between GC3-bias and protein expression (*Horn, 2008*). However, the hypothesis that GC3 codons increase expression was not addressed experimentally. To test this hypothesis, we cloned wild-type and (human) codon-optimised (GC3-bias is also observed in human sequences) *Gaussia* luciferase (g*LUC*) genes (*Shao and Bock, 2008*) in a *T. brucei* inducible expression construct (*Figure 1A*). Importantly, we used a construct that integrates reproducibly as a single copy at a single site in the genome, eliminating position effects that arise due to integration at different genomic sites (*Alsford et al., 2005*). All other sequences, except for the protein coding sequence, remained constant for each experiment performed with these constructs. The relative abundance of GC3 codons, typically found in highly expressed *T. brucei* genes (*Horn, 2008*), is illustrated in heat-map format (*Figure 1A*). We selected g*LUC* because the expression of the gLUC protein can be assessed over a wide dynamic range using an activity assay, and inducible expression was employed so that we could determine whether gLUC expression was toxic to *T. brucei*. We assembled bloodstream form *T. brucei* strains with each construct integrated into the genome and first assessed gLUC expression on protein blots. We observed robust inducible expression and relative induced expression levels that were as predicted by our hypothesis, revealing 6-fold higher luciferase expression from the GC3-enriched gene as determined by densitometry (*Figure 1B*); three independent clones using each construct revealed similar results and we saw no evidence of toxicity.

Encouraged by these initial results, we assembled constitutive expression constructs containing three new synthetic g*LUC* genes (*Supplementary File 1*, sheet 1) with the maximal or minimal number of GC3 codons or with alternating GC3 and AT3 codons; again illustrated in heat-map format (*Figure 1C*). *T. brucei* strains expressing these constructs were assessed by luciferase activity assay, which again revealed relative expression levels that were significantly different and as predicted by our hypothesis (*Figure 1C*). We observed 4-fold higher expression from the high-GC3 gene relative to the medium-GC3 gene and strikingly undetectable expression from the low-GC3 gene; in each case, a pair of independent clones derived using each construct revealed similar expression levels. These results indicate that codon usage has the potential to account for the full dynamic range of protein expression levels observed in trypanosomes.

The Green Fluorescent Protein is often used as a reporter or as a protein tag for gene expression and subcellular localisation studies, and this is also the case in trypanosomatids. We, therefore, used *GFP* to test the impact of codon usage. As above, we assembled expression constructs containing three synthetic *GFP* genes (*Supplementary File 1*, sheet 1) with the maximal or minimal number of GC3 codons or with alternating GC3 and AT3 codons; see heat-maps (*Figure 1D*). *T. brucei* strains expressing these constructs were assessed by quantitative protein blotting, which once again revealed the expected relative expression levels (*Figure 1D*), further validating our hypothesis. Expression from the high-GC3 *GFP* gene was higher than from the medium-GC3 gene and, as in the case of gLUC above, expression of the low-GC3 gene was undetectable; again, pairs of independent clones using each construct revealed similar expression levels.

### GC3 codon bias increases mRNA expression

We next assessed mRNA abundance in the strains detailed above. Initially, we used a tubulin (*TUB*) fragment as a probe for the wild-type and human codon-optimised g*LUC* mRNAs, since the untranslated regions in these reporter constructs are from a *TUB* gene. This allowed us to use native *TUB*

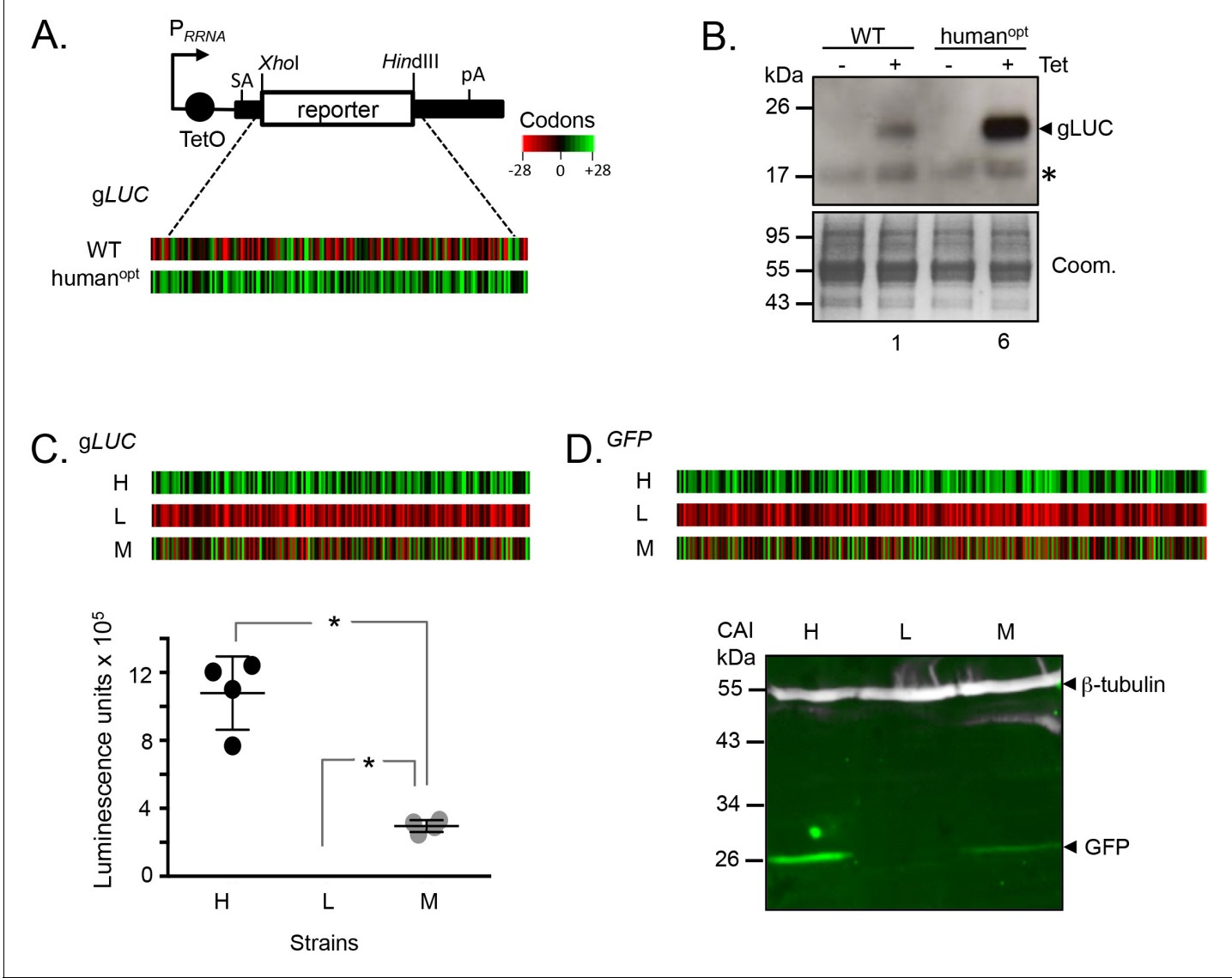

**Figure 1.** Protein expression is increased by GC3 codons in *T.brucei*. (**A**) Schematic map of the pRPa^i^-based, tetracycline-inducible reporter construct. Relevant restriction sites are shown. Black bars, *tubulin* untranslated regions; arrow, pol-I promoter; pA, polyadenylation site; SA, splice-acceptor site. The heat-maps of the wild-type and human codon optimised *gLUC* genes indicate level (percentage) of codon over-representation (green) and under-representation (red) in highly expressed genes. (**B**) Protein blot analysis of *gLUC* expression in *T. brucei*. *, cross-reactive band. The Coomassie-stained panel serves as a loading control; the strong band at approximately 55 kDa is the abundant Variant Surface Glycoprotein (VSG). The numbers indicate proportional luciferase expression, based on densitometry. Three independent clones gave similar results for each construct. (**C**) The heat-maps of synthetic *gLUC* reporter genes indicate codon usage as in A above. The plot indicates luciferase activity for each reporter in *T. brucei*; four readings from two independent strains. Error bars, standard deviation. *, p<0.0001; one-way ANOVA test. (**D**) The heat-maps of synthetic *GFP* reporter genes indicate codon usage as in A above. The LICOR protein blot indicates GFP expression for each reporter in *T. brucei*; β-tubulin serves as a loading control. Two independent clones gave similar results for each construct.

DOI: https://doi.org/10.7554/eLife.32496.003

transcripts as a loading control and, since the *gLUC* gene is short, to identify *gLUC* transcripts migrating below native *TUB* transcripts on the blot. Analysis of *gLUC* RNA extracted from the *T. brucei* strains shown in *Figure 1A* revealed increased expression from the GC3-enriched gene (*Figure 2A*). These results indicated that mRNA expression was also increased by GC3 codons.

To further test the link between GC3 codons and mRNA abundance, we assessed expression of the synthetic *gLUC* and *GFP* genes described above (*Figure 1C,D*). In this case, we engineered a unique λ-phage DNA sequence immediately downstream of each protein-coding sequence such that

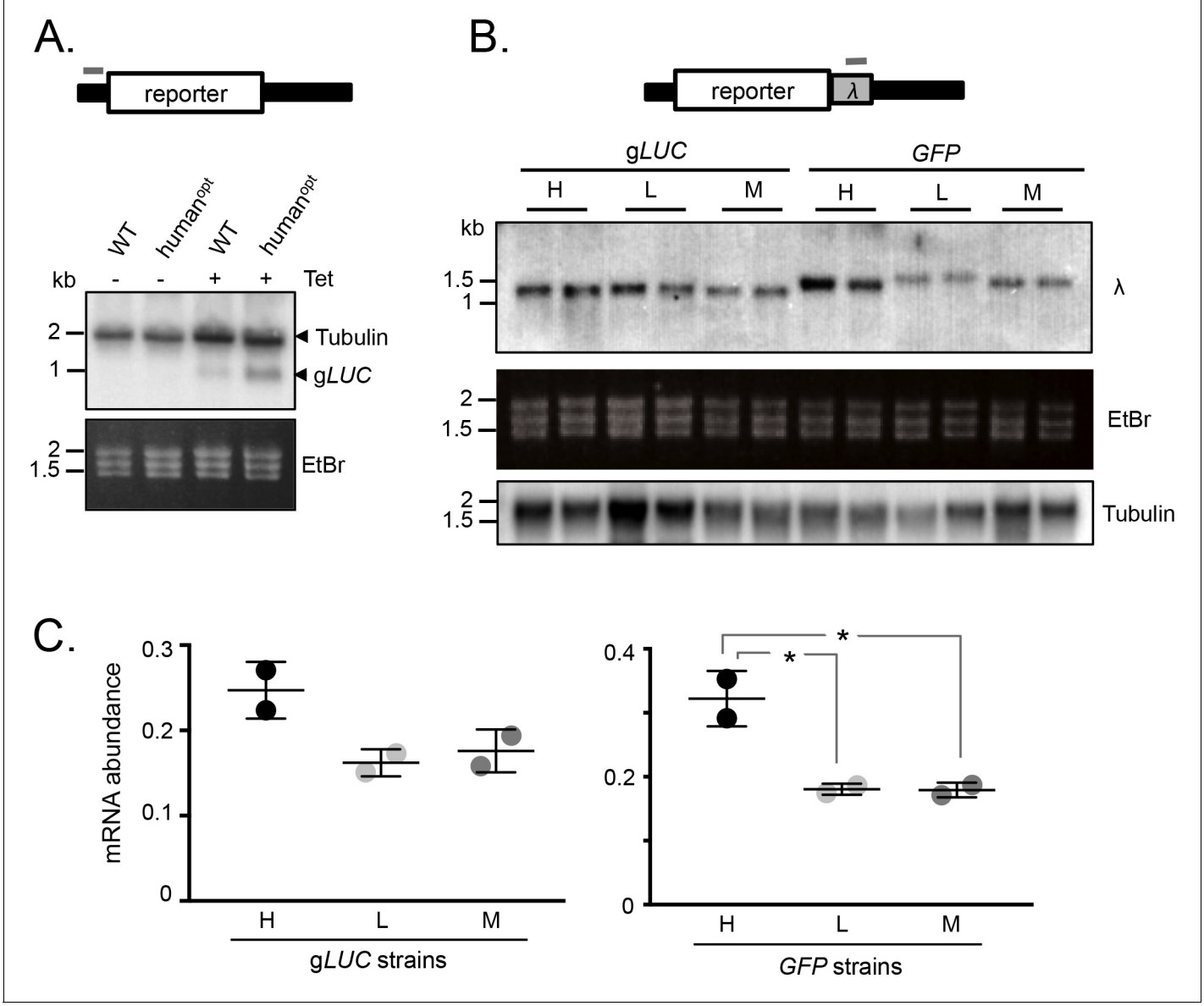

**Figure 2.** mRNA expression is increased by GC3 codons in *T.brucei*. (A) Schematic map of the reporter cassette. The grey bar indicates the position of the *tubulin* untranslated region probe. The RNA blot indicates native *tubulin* transcripts and the *gLUC* transcripts. An ethidium bromide stained gel serves as an additional loading control. (B) Schematic of the reporter cassette incorporating a *lambda* 3'-untranslated segment. The grey bar indicates the position of the *lambda* untranslated region probe. The upper RNA blot shows *gLUC* and *GFP* transcripts. An ethidium bromide stained gel and a replicate blot probed for *tubulin* serve as loading controls. Pairs of independent strains were analysed for each reporter construct. (C) Phosphorimager-based quantification of reporter expression in B. Error bars, standard deviation from two independent strains. Values were corrected for loading (*tubulin*). *, p<0.02; one-way ANOVA test.

DOI: https://doi.org/10.7554/eLife.32496.004

it would be included in the 3'-untranslated region (*Figure 2B*). *T. brucei* strains expressing these reporter constructs were assessed on RNA blots using a unique 'λ−probe' (*Figure 2B-C*) and an equivalent blot was probed with *TUB* as a loading control. We observed higher mRNA expression from the high-GC3 relative to the medium-GC3 or low-GC3 *gLUC* and *GFP* genes; the difference was <2 fold for *gLUC* but approximately 2-fold for *GFP* and achieved statistical significance in the latter case. Notably, low-GC3 *gLUC* and *GFP* genes yielded slower-migrating transcripts (*Figure 2B*), which may reflect longer poly(A) tails, as also observed for poorly translated mRNAs with a low proportion of optimal codons in other eukaryotes (*Lima et al., 2017*). These results

indicate that codon usage does indeed impact mRNA expression in *T. brucei*. While the range of values reaches almost 2-fold differential for mRNA (*Figure 2C*), the range is substantially greater for protein (*Figure 1C*) pointing to an impact on translation that quantitatively exceeds the impact on mRNA levels.

## Genome scale analysis of codon usage bias

Having established that GC3-codons increase mRNA and protein expression, we next assessed codon usage at the genomic scale (*Figure 3A*). The average Codon Adaptation Index (CAI) (*Sharp and Li, 1987*) for the full set of >7,000 *T. brucei* genes is 0.712 (*Figure 3A*), with a maximal CAI of 0.883 for the α-tubulin genes. The reference set of highly expressed genes (see Materials and methods) has an average CAI of 0.782 (*Figure 3A*) and, consistent with the view that tandem amplification is another mechanism used by trypanosomatids to increase gene expression (*Horn, 2008*), there are many 'tandem genes' in this reference set. Importantly, the set of genes encoding detected proteins in a published proteome dataset (*Urbaniak et al., 2012*), had a substantially higher average CAI (0.724), relative to the proteins that were not detected (0.701, *Figure 3A*). Just over 500 genes encoding abundant proteins detected in a *Leishmania mexicana* proteome study were also enriched for high CAI values (*Paape et al., 2008*).

When we assessed CAI in relation to location on trypanosomatid chromosomes; we found that many genes immediately adjacent to divergent or convergent polycistronic transcription initiation or termination regions displayed relatively low CAI values (*Figure 3A,B*); the full cohort of 343 of these 'strand-switch' genes yielded an average CAI of 0.685 (*Figure 3A*) and *T. brucei* chromosome 3 is shown to illustrate (*Figure 3B*). This reflects the previous observation of base-skew asymmetry flanking these regions (*Nilsson and Andersson, 2005*). We also assessed codon usage at the genomic scale in *Leishmania major* and similarly found that tandem paralogous genes and genes at

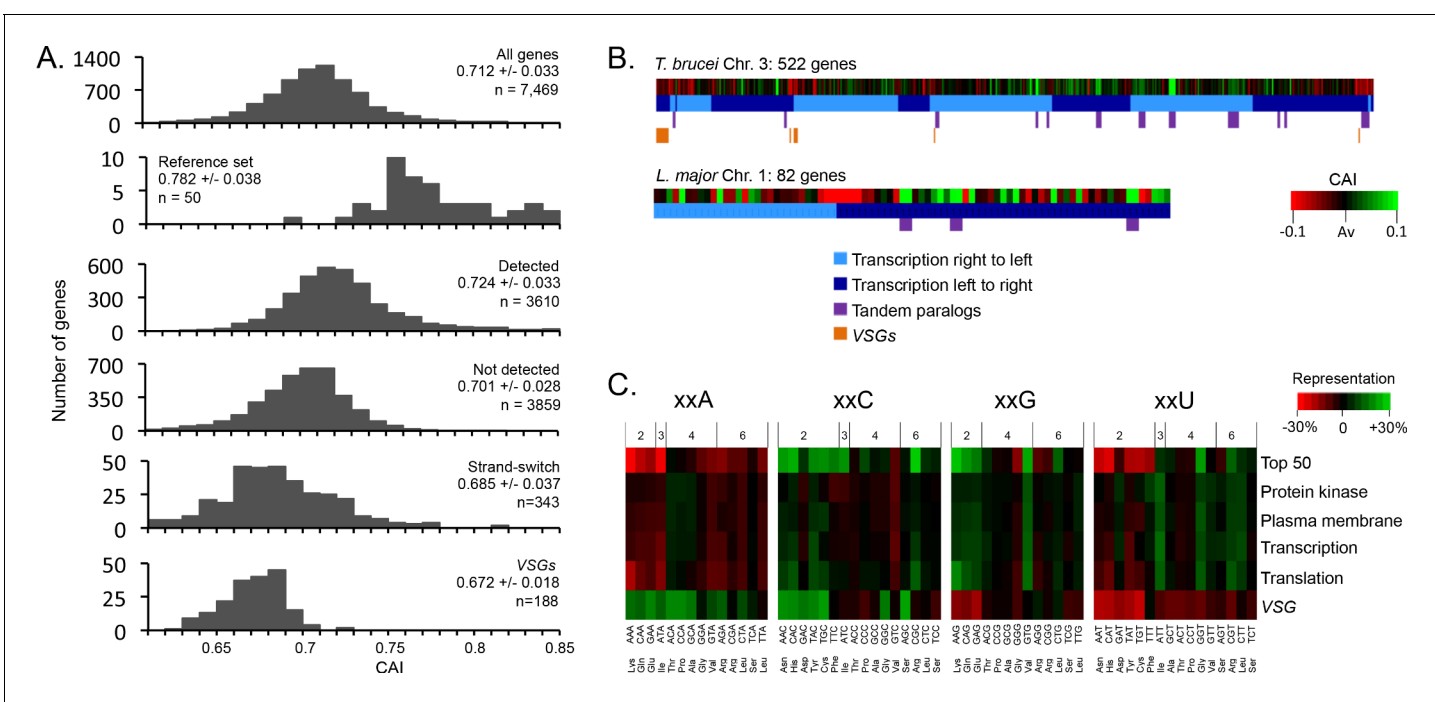

**Figure 3.** Genome scale analysis of codon usage bias. (**A**) CAI value distribution is shown for all non-redundant *T. brucei* genes and the cohorts of genes indicated. See the text for more detail on each cohort. (**B**) CAI values are shown in heat-map format (deviation from average, Av) on physical maps of *T. brucei* chromosome 3 and *L. major* chromosome 1. Salient features are indicated. (**C**) Codon representation (third position difference), relative to the average usage across the genome, is shown within the cohorts of *T. brucei* genes indicated; protein kinase activity (GO:0004672), plasma membrane (GO:0005886), transcription (GO:0006350), translation (GO:0006412). The numbers above the heat-map indicate the number of redundant codons available in each case.

DOI: https://doi.org/10.7554/eLife.32496.005

transcription switch regions have inflated and deflated CAI values, respectively; *L. major* chromosome 1 is shown to illustrate this (*Figure 3B*).

An analysis of codon representation in the reference set of highly expressed genes revealed the expected over-representation of GC3 codons, which is particularly pronounced when there are only two alternative codons for an amino acid (*Figure 3C*). Indeed, in the case of xxC/T wobble codons the xxC rather than xxT version of the tRNA is always present for these amino acids (Asn, Asp, Cys, His, Phe, Tyr) (*Horn, 2008*). In contrast, when there are 3–6 alternative codons, the xxT version of the wobble tRNA is typically present (Ile, Ala, Pro, Thr, Val, Leu, Arg, Ser$^{TCT}$) and both alternative wobble codons are often over-represented in these cases (Ile, Ala, Leu, Arg, Ser$^{TCT}$).

Variant Surface Glycoprotein (VSG) genes in *T. brucei*, which are involved in immune evasion and are subject to allelic exclusion, display unusual codon usage bias (*Alvarez et al., 1994*; *Chanda et al., 2007*; *Cross et al., 2014*). Distinct from highly expressed and indeed other gene families, A3 codons are over-represented, while G3 codons, and notably almost all U3 codons, are under-represented in the *VSG* genes (*Figure 3C*). Consistent with this, we find that the cohort of 188 intact *VSGs* display low CAI values (*Figure 3A,B*), with an average of 0.672. The single expressed *VSG* produces the most abundant mRNA and protein in bloodstream form *T. brucei* but a high rate of translation is probably not required due to a high rate of transcription by RNA pol-I, combined with remarkably slow turnover of both the *VSG* mRNA (half-life ~4.5 hr) (*Ehlers et al., 1987*) and protein (half-life ~33 + /- 9 hr) (*Seyfang et al., 1990*); the mean mRNA half-life in bloodstream-form *T. brucei* is 16 min (*Fadda et al., 2014*). Indeed, sub-optimal translation, due to low G3 and high A3-usage in *VSGs*, may be necessary to prevent these multi-copy genes, distributed throughout the genome, from compromising *VSG* allelic exclusion and the immune evasion strategy of bloodstream form *T. brucei*. This particular virulence strategy could fail if constitutive pol-II transcription yielded sufficient VSG at the cell surface to simultaneously present multiple variants to the host.

Since codon-pair bias is observed in other genomes (*Quax et al., 2015*), and some pair-bias combinations have the potential to be in conflict with increased codon optimality, we also examined these features of *T. brucei* protein-coding sequences. This revealed enrichment of duplicated codons (*Figure 4A*) and other examples of pair-bias (*Figure 4B,C*). Codon co-occurrence is thought to facilitate tRNA recharging, while the over-represented (examples include xxC/Axx, xxU/Gxx, xxC/xxC and xxG/xxG) and under-represented (examples include xxA/Gxx, xxU/Axx, xxC/xxG, xxG/xxC) pairs

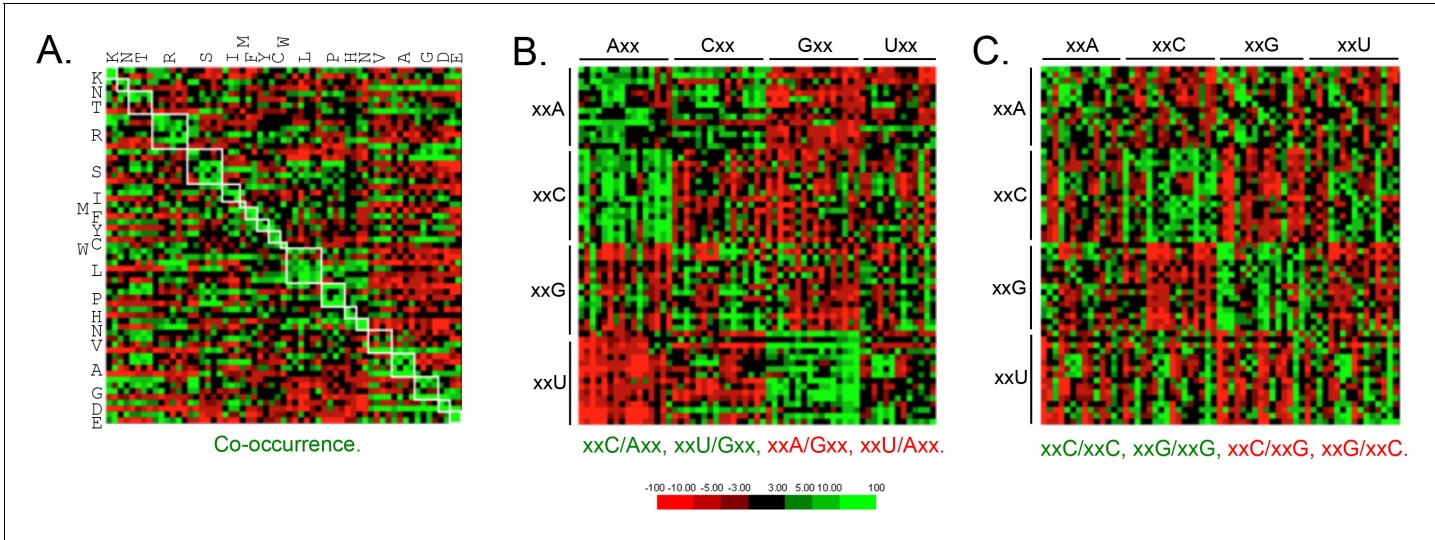

**Figure 4.** Genome scale analysis of codon pair bias in *T. brucei*. (**A**) Codon co-occurrence by encoded amino acid. Amino acid pairs are over-represented; highlighted by white boxes. (**B**) Analysis of third position followed by first position pairs. Examples of over-represented pairs are shown in green and examples of under-represented pairs are shown in red. (**C**) Analysis of third position and third position pairs. Examples are shown as in B. Amino acids and codons on the vertical axis precede those on the horizontal axis.

DOI: https://doi.org/10.7554/eLife.32496.006

(*Figure 4B,C*) are thought to reflect more optimal tRNA interactions at the A and P sites that impact translation fidelity (*Quax et al., 2015*). We are not aware of evidence supporting a direct impact of codon pair-bias on mRNA or protein expression, so these differences were not considered further here. We do note, however, that an over-representation of xxU/Gxx pairs, and under-representation of xxC/xxG and xxG/xxC pairs, limits the accumulation of 'optimal' GC3 codons.

## Predicting relative mRNA and protein expression using sequence alone

The results above suggested that it might be possible to predict relative mRNA and protein expression in trypanosomatids, based on protein coding-sequences alone. To address this question, we generated a new *T. brucei* transcriptome dataset. This dataset comprised three replicates, was of sufficient depth to provide >114 million mapped reads, and yielded pair-wise Pearson correlation coefficients of >0.999 (*Figure 5—figure supplement 1*). This depth of coverage and correspondence provides an excellent level of resolution for the *T. brucei* transcriptome. We analysed this dataset alongside a published *T. brucei* proteome dataset (*Urbaniak et al., 2012*). An initial comparison of mRNA and peptide expression yielded a Pearson's correlation coefficient of 0.57 (*Figure 5A*, *Supplementary file 1*, sheet 2). Thus, as in other cell types (*Ly et al., 2014*), mRNA abundance is predictive of protein abundance. This relationship displays the best fit with a logarithmic trend-line, whereas the best fit is with a linear trend-line when only 10% of the most highly expressed mRNAs are excluded from the analysis. This may reflect poor translation of highly abundant mRNA's, as noted previously for transcripts encoding ribosomal proteins (*Antwi et al., 2016*), but might equally reflect experimental bias due to limitations in the effective dynamic range of proteome analysis (*Urbaniak et al., 2012*).

Short genes produce relatively more mRNA in *T. brucei* (*Fadda et al., 2014*; *Jha et al., 2014*), so we also considered gene length for our analysis. Indeed, we found that gene length is inversely correlated with both mRNA expression in our dataset (*Figure 5B*, *Supplementary file 1*, sheet 2) and with protein expression (*Figure 5C*, *Supplementary file 1*, sheet 2) and that long genes tend to make less mRNA and protein.

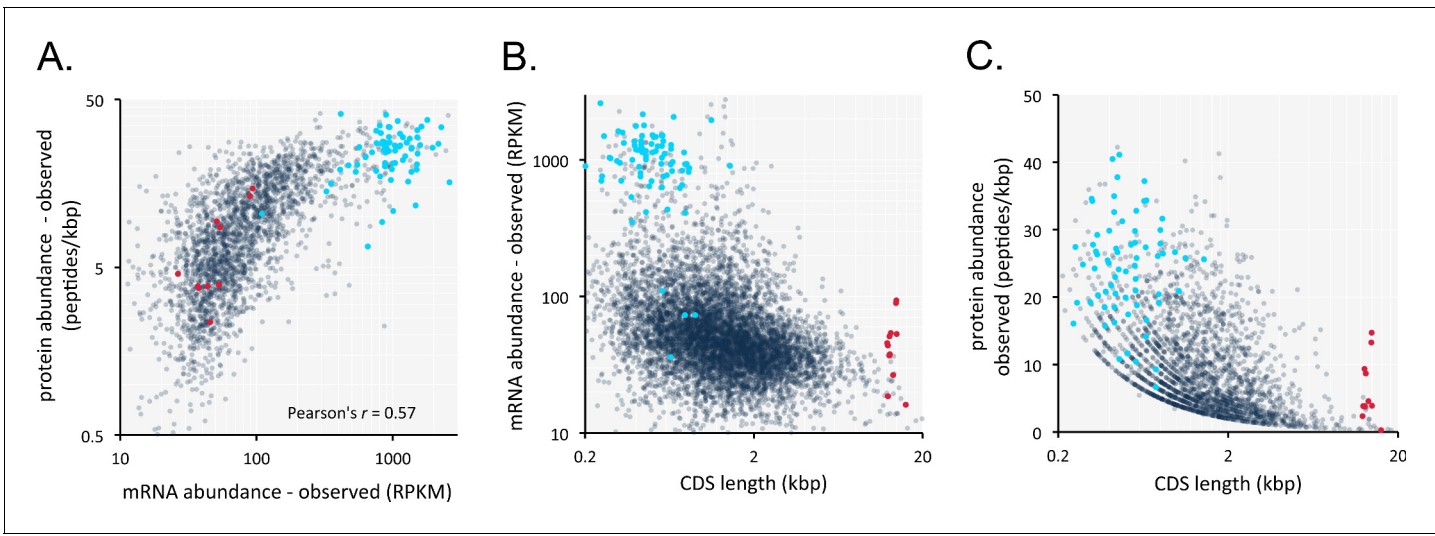

**Figure 5.** Transcriptome and proteome data and the impact of gene length in *T.brucei*. (A) Correspondence between observed mRNA and protein expression. (B) Relationship between observed mRNA expression and protein coding sequence (CDS) length. RPKM, Reads Per Kilobase of transcript per Million mapped reads. (C) Relationship between observed protein expression and protein coding sequence (CDS) length. Cohorts of particularly long (red, 13.4 ± 1 kbp, n = 11) and short (blue, 0.55 ± 0.22 kbp, n = 67) genes, encoding dynein heavy chains and ribosomal proteins, respectively, are highlighted. n = 2315 genes for panels A and C, n = 7225 genes for panel B.

DOI: https://doi.org/10.7554/eLife.32496.007

The following figure supplement is available for figure 5:

**Figure supplement 1.** RNA-seq data.

DOI: https://doi.org/10.7554/eLife.32496.008

An analysis of the correspondence between codon usage and observed *T. brucei* mRNA expression (n = 7225) yielded a Pearson correlation coefficient of 0.48 (*Figure 6A*, *Supplementary file 1*, sheet 2). We found this quite remarkable, that a measure of mRNA abundance can be predicted based on the coding-sequence alone. Since shorter coding-sequences yield more mRNA (*Figure 5B*), we derived a simple formula including a penalty for coding-sequence length [CAI - (0.03 x $\sqrt{}$coding sequence length in kbp)]. This penalty improved the correspondence between CAI-values and observed mRNA expression and yielded a correlation coefficient of 0.52 (*Figure 6B*, *Supplementary file 1*, sheet 2); using this approach, >70% of our estimates of relative expression are within 2-fold of the observed values (*Figure 6C*).

Our RNA-seq data above are derived from the bloodstream-form life-cycle stage of *T. brucei*. The expression of some mRNAs and proteins is developmentally regulated, however. We, therefore, analysed our previously published RNA-seq data from both bloodstream-form and insect-form cells (*Hutchinson et al., 2016*). Using these data, minus those 2.8% of genes that differed >3 fold

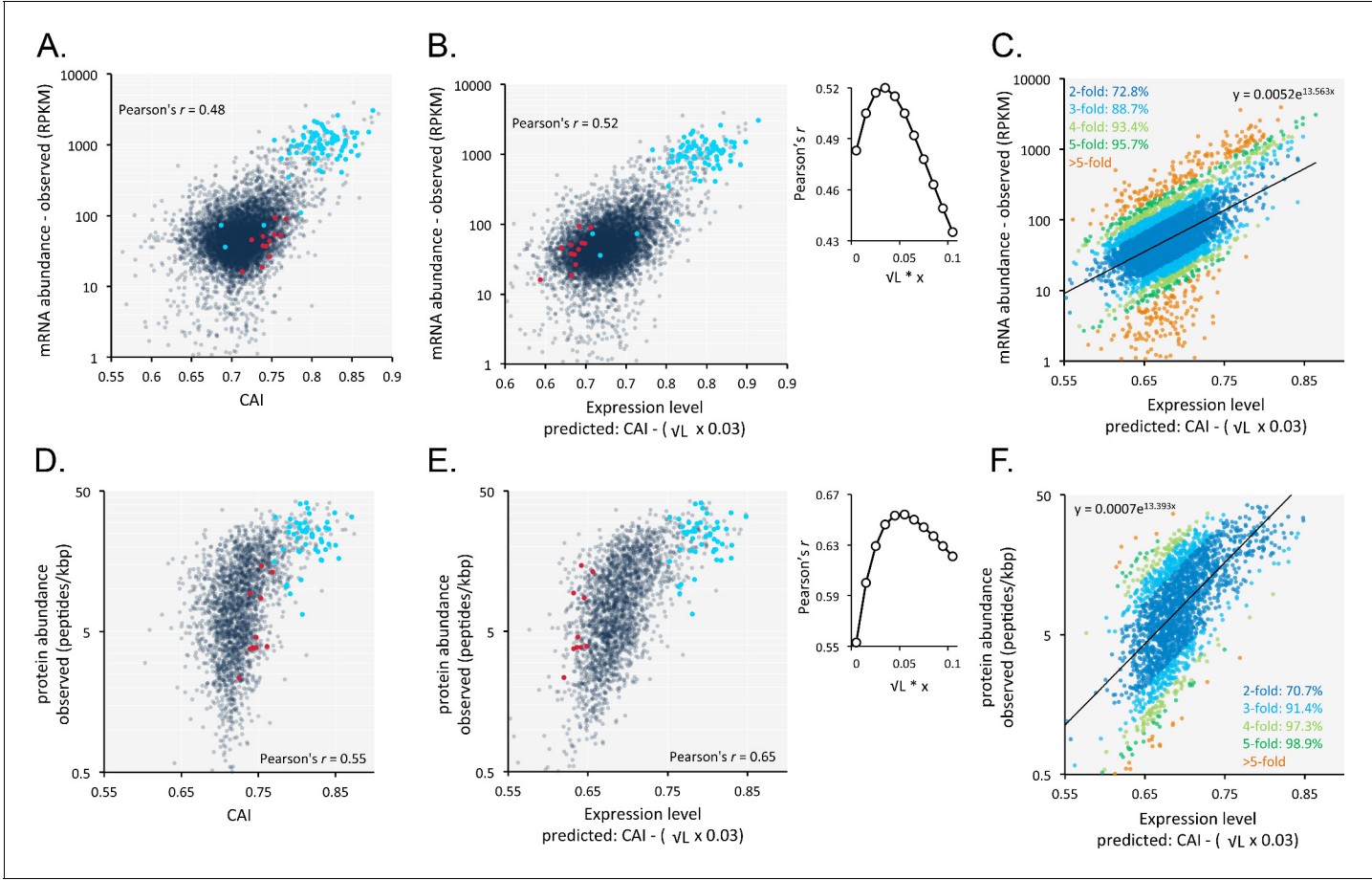

**Figure 6.** Codon usage is predictive of relative mRNA and protein expression in *T.brucei*. (**A**) Correspondence between relative observed mRNA expression and CAI. (**B**) Correspondence between relative observed mRNA levels and predicted expression based on CAI and CDS length in kbp (L); the inset shows the impact of length-correction on the correlation coefficient. RPKM, Reads Per Kilobase of transcript per Million mapped reads. (**C**) As in B but showing proportions of expression measures within 2- to 5-fold of the predictions; the formula for the exponential trend-line is indicated. (**D**) Correspondence between relative observed protein expression and CAI. (**E**) Correspondence between relative observed protein levels and predicted expression based on CAI and CDS length in kbp (L); inset as in B above. (**F**) As in E but showing proportions of expression measures within 2- to 5-fold of the predictions; the formula for the exponential trend-line is indicated. A-B, D-E; Cohorts of particularly long (red) and short (blue) genes (see *Figure 5*) are highlighted. n = 7225 genes for panels A-C, n = 2315 proteins for panels D-F.

DOI: https://doi.org/10.7554/eLife.32496.009

The following figure supplement is available for figure 6:

**Figure supplement 1.** Length-adjusted CAI is predictive of relative mRNA expression in previously published datasets.

DOI: https://doi.org/10.7554/eLife.32496.010

between stages (201 of 7191 genes), we once again observed correspondence between length-adjusted CAI values and RNA-seq expression data (*Figure 6—figure supplement 1A,B*). Pearson correlation coefficients were 0.54 and 0.53 for the bloodstream-form (*Figure 6—figure supplement 1A*) and insect-form data (*Figure 6—figure supplement 1B*), respectively (improved by 5.5% and 4.6% by the length-adjustment). RNA-seq data from insect-form cells generated by another research group (*Christiano et al., 2017*) also revealed correspondence with length-adjusted CAI values (*Figure 6—figure supplement 1C*). The Pearson correlation coefficient was 0.5 in this case (improved by 3% by the length-adjustment). Thus, our findings apply to multiple *T. brucei* life cycle stages and to independently generated datasets.

A similar analysis of the impact of codon usage on the proteome of *T. brucei* (n = 2315) also revealed a substantially improved correlation when we applied the same penalty for coding-sequence length. In this case, the correlation between CAI values and observed protein expression yielded a correlation coefficient of 0.55 (*Figure 6D*, *Supplementary file 1*, sheet 2), which increased to 0.65 following the length-adjustment (*Figure 6E*, *Supplementary file 1*, sheet 2); again,>70% of our estimates of relative expression are within 2-fold of the observed values (*Figure 6F*).

## Coding sequences predict the expression of protein cohorts and complexes

Predictions of 'steady-state' abundance for individual proteins are prone to 'under-sampling' errors in proteome data. Since many proteins function in multi-component complexes or share similar functions with related proteins, we analysed cohorts of genes encoding components of protein complexes or related functions (*Supplementary file 1*, sheet 3). For these cohorts (n = 23), codon usage-based predictions, when compared to observed measures, yielded a striking Pearson correlation coefficient of 0.84 (*Figure 7*). This further reinforces the view that codon usage bias plays a major role in controlling protein expression.

Using our relative expression estimates, we derived values for numbers of both mRNA and protein molecules per bloodstream form cell (*Supplementary File 1*, sheet 2). These are based on 19,000 non-*VSG* mRNAs (*Haanstra et al., 2008*) and 100 million non-VSG proteins per cell, similar to estimates for yeast (*Milo, 2013*). We also derived adjusted protein estimates based on the mRNA levels we observe, which provides a proxy adjustment for gene copy-number. There are limitations of course; including expected inflated values for cell cycle specific proteins or developmentally regulated proteins. Indeed, we see examples of such complexes in *Figure 7*; the abundance of the cell cycle specific kinetochore complex (*Akiyoshi and Gull, 2014*) and the bloodstream down-regulated editosome complex (*Stuart et al., 2005*) are 3.7-fold and 2.7-fold below the predictions based on the trend-line, respectively (*Figure 7*). Notably, the analyses shown in *Figure 7* also appears to reflect a particular technical limitation typically seen when using peptide-based proteomic strategies, the under-representation of membrane–associated proteins; the abundance of the o-sector of the V-ATPase (*Baker et al., 2015*) is 2.5-fold below the prediction on this plot. Despite the limitations, we believe our estimates of protein expression levels and stoichiometry will be of value for expression of recombinant proteins in trypanosomatids (*Fritsche et al., 2007*) and for understanding trypanosomatid biology.

## Codon usage corresponds with ribosome occupancy and mRNA stability

Favoured or optimal codons may increase protein expression by increasing the translation rate. To explore this possibility, we compared our predicted expression values with published ribosome profiling data (*Vasquez et al., 2014*) and observed correspondence with measures of translation efficiency (*Figure 8A*); the correlation coefficient was 0.35 (improved by 6.8% by the length-adjustment). Optimal codons may similarly increase mRNA abundance, if ribosome interactions stabilise mRNAs. Indeed, a comparison between CAI values and published *T. brucei* mRNA half-life data (*Fadda et al., 2014*) revealed a correspondence (*Figure 8B*); the correlation coefficient was 0.37. Notably, correspondence decreased (3.2% in bloodstream-form cells; or improved only 0.3% in insect-stage cells) when the length-adjustment was applied in this case. This supports the view that codon usage bias controls mRNA stability in a length-independent manner. We suggest that optimal

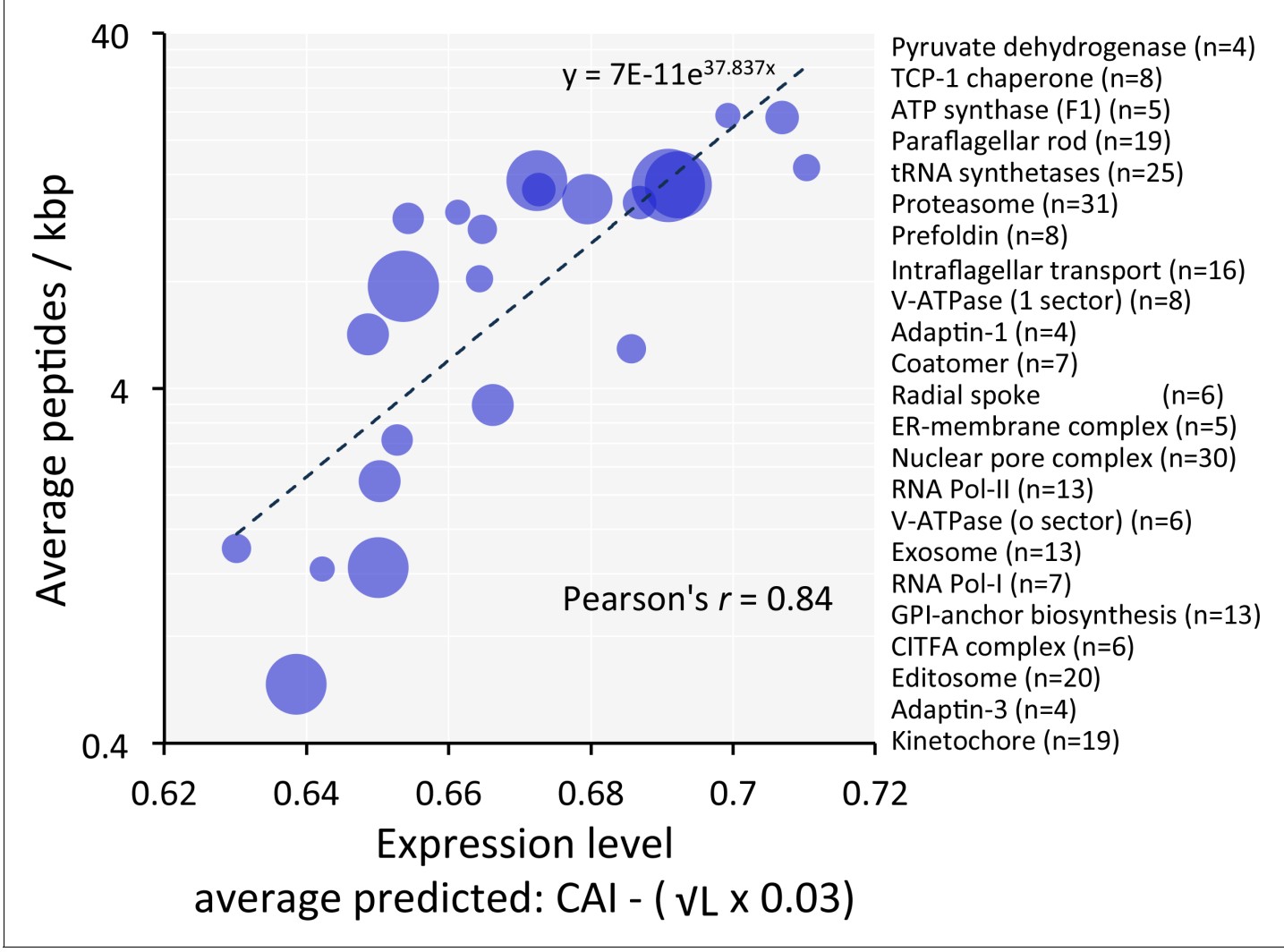

**Figure 7.** Codon usage predicts the relative expression of protein complexes and cohorts of proteins with related functions in *T.brucei*. Correspondence between observed peptide counts and predicted abundance based on CAI. The complexes and cohorts are listed in order of peptides/kbp and number of proteins is indicated for each; protein numbers are also reflected by the symbol sizes. The formula for the exponential trend-line is indicated. n = 23 cohorts, n = 277 proteins.

DOI: https://doi.org/10.7554/eLife.32496.011

codons increase ribosome occupancy and translation, which in turn protects mRNA from (length-independent) decapping/deadenylation, thereby also increasing mRNA stability.

## Predicting mRNA and protein expression in other trypanosomatids

We suspected that codon usage bias also has a substantial impact on both mRNA and protein expression in other trypanosomatids. We, therefore, analysed previously published RNA-seq and proteomic data from *Trypanosoma vivax*, another African trypanosome that causes disease in cattle and other livestock (*Jackson et al., 2015*). Using these data, we observed correspondence between length-adjusted CAI values and both RNA-seq (*Figure 9A*) and proteomic (*Figure 9B*) expression data. The Pearson correlation coefficients were 0.47 and 0.63, respectively (improved by 2.1% and 10% by the length-adjustment). This was also the case (*Figure 9C*) when we analysed RNA-seq data from *Leishmania mexicana* (*Fiebig et al., 2015*), a trypanosomatid parasite thought to have diverged from the African trypanosome lineage >120 million years ago (*Harkins et al., 2016*); the correlation coefficient was 0.43 (improved by 2.4% by the length-adjustment). These findings support the view

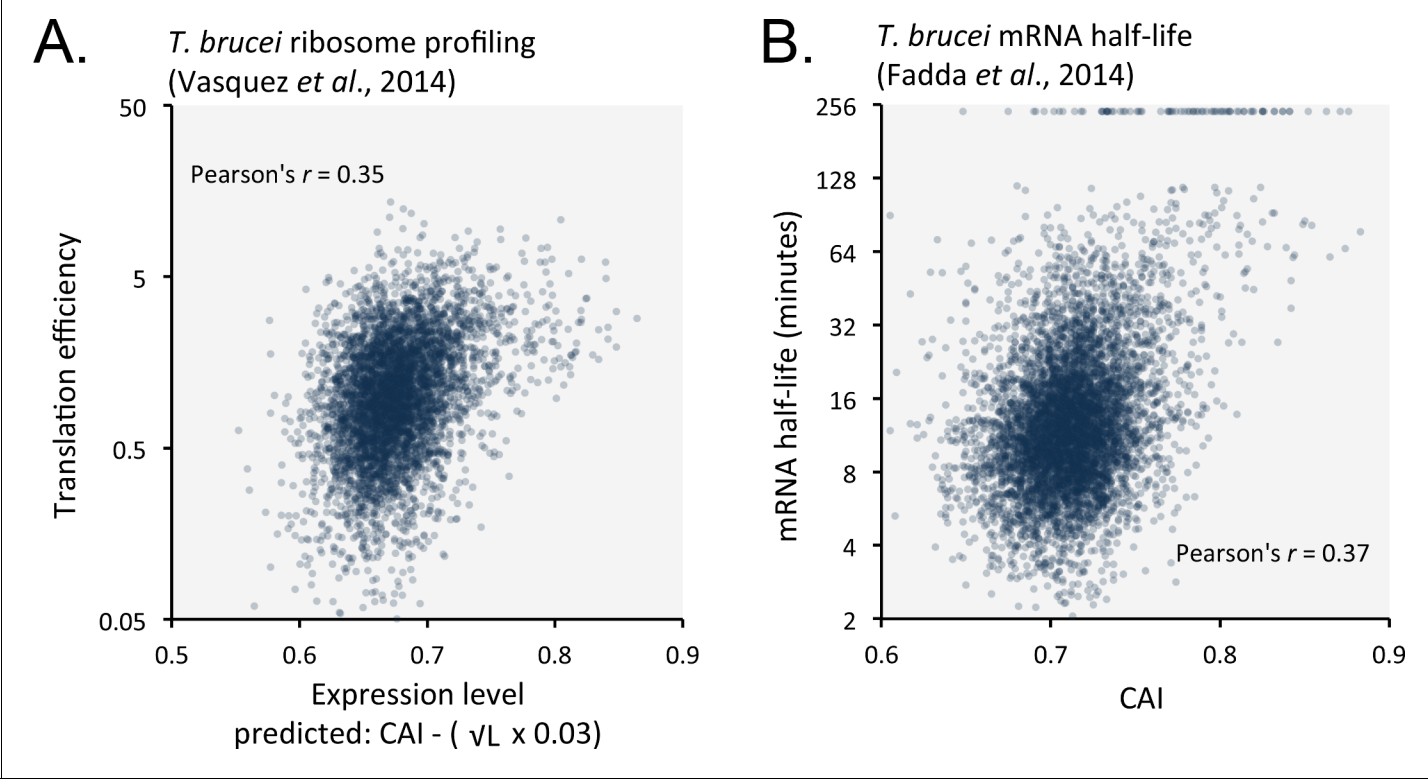

**Figure 8.** Length-adjusted CAI and CAI are predictive of translation efficiency and mRNA half-life, respectively, in previously published data from *T. brucei*; the data source is indicated in each case. (**A**) Correspondence between translation efficiency (footprint levels/mRNA levels) and length-adjusted CAI. n = 4880 genes. Data from bloodstream-form cells is shown; correlation coefficient for insect-form cells was 0.36 (improved by 3.3% by the length-adjustment). (**B**) Correspondence between mRNA half-life and CAI. n = 6333 genes. Data from bloodstream-form cells is shown; correlation coefficient for insect-form cells was 0.42.

DOI: https://doi.org/10.7554/eLife.32496.012

that codon usage bias, and gene length, control mRNA and protein expression in many, if not all, trypanosomatids.

Taken together, our results illustrate a remarkable ability to predict mRNA and protein expression and abundance at a transcriptomic and proteomic scale based on protein-coding sequences alone. Indeed, relative expression can now be predicted for the full complement of proteins, including thousands of proteins that fail to register in current proteome datasets (*Supplementary file 1*, sheet 2). To facilitate further predictions we provide extended sets of CAI values for *T. brucei* (*Supplementary file 1*, sheet 4), *T. vivax* (*Supplementary file 1*, sheet 5) and *L. mexicana* (*Supplementary file 1*, sheet 6).

## Discussion

Cells have evolved mechanisms to coordinate the expression of a large number of proteins of widely differing abundance. Codon usage contributes to gene expression control but it can be challenging to investigate the impact of codon usage bias at a genomic and proteomic scale in most eukaryotes because gene expression control operates at many levels, through transcription control in particular. Trypanosomatids display constitutive RNA pol-II transcription and, therefore, rely heavily upon post-transcriptional control, consequently presenting excellent models for the study of mRNA stability and translation control. It was previously suggested that codon usage and corresponding tRNA gene dosage plays an important role in gene expression control in trypanosomatids (*Horn, 2008*) but experimental tests of this translational selection hypothesis were lacking. We now show that

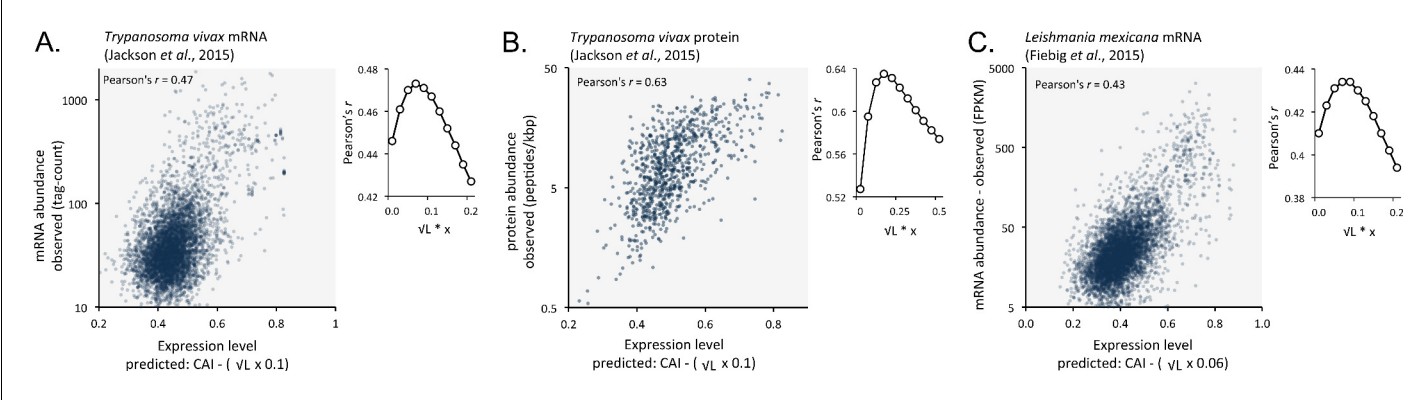

**Figure 9.** Length-adjusted CAI is predictive of relative mRNA and protein expression in previously published data from the other trypanosomatids, *T. vivax* and *Leishmania mexicana*; the data source is indicated in each case. The plots indicate correspondence between relative observed mRNA or protein expression and our predictions based on CAI and CDS length in kbp (L). (A) *T. vivax* mRNA expression. n = 5170 genes. (B) *T. vivax* protein expression. n = 859 proteins. (C). *Leishmania mexicana* mRNA expression. n = 5715 genes. The insets show the impact of length-correction on the correlation coefficient. FPKM, Fragments Per Kilobase of transcript per Million mapped reads.

DOI: https://doi.org/10.7554/eLife.32496.013

codon usage has a major impact on gene expression in *T. brucei* and that both relative mRNA and protein expression can be predicted based on coding sequence alone.

Using two distinct reporter genes in bloodstream form *T. brucei*, we show that codon usage has the capacity to control relative protein abundance over several orders of magnitude. Although to a lesser extent, mRNA abundance was also dependent upon codon usage. Indeed, codon optimality increases mRNA stability in yeast (*Presnyak et al., 2015*), by reducing Dhh1p-dependent decay (*Radhakrishnan et al., 2016*). Anticipating a major impact of codon bias on gene expression in *T. brucei*, we predicted relative mRNA and protein expression at a global scale using coding sequences alone. We found that codon usage was indeed predictive of observed mRNA and protein expression and report correspondence between the relative abundance of thousands of native proteins and thousands of native mRNA transcripts. We find it quite remarkable that our predictions of steady-state mRNA and protein expression, based only on coding sequences, correlate with observed abundance data; even taking no account of controls mediated by untranslated sequences or differences in protein turnover.

Under-sampling is one of the major current challenges in terms of determining relative protein abundance in cells and tissues (*Ly et al., 2014*). With this in mind, we predicted protein expression for cohorts of genes that encode components of protein complexes or related functions. We found a remarkable correlation between our sequence-based predictions and observed protein levels. Notably, for proteins that form multi-component complexes, protein turnover is likely dependent upon complex assembly, with destabilisation of unassembled subunits.

In addition to codon usage, gene length also has an impact on mRNA and protein expression, and this is proposed to be due to increased turnover of longer transcripts (*Fadda et al., 2014*; *Jha et al., 2014*). This is entirely feasible because individual lesions in mRNA will reduce abundance and lesions may be proportional to length. When we applied a gene-length adjustment, this failed to improve the correspondence between codon usage and mRNA half-life, consistent with the idea suggested previously (*Fadda et al., 2014*) that transcript length has a greater impact on co-transcriptional degradation rather than on co-translational degradation. Perhaps the former is dominated by (length-dependent) endonuclease activity while the latter is dominated by (length-independent) exonuclease activity, following decapping and deadenylation.

Ribosome profiling, nascent chain profiling (*Chadani et al., 2016*) and analysis of newly synthesised proteins (*Zur et al., 2016*) are powerful methods for probing translational efficiency and translational pausing. The former approach has been applied to *T. brucei* (*Vasquez et al., 2014*). Our analysis of codon usage in relation to these translation efficiency data, as well as mRNA half-life data (*Fadda et al., 2014*), also from *T. brucei*, suggested that optimal codons increase ribosome

occupancy and mRNA stability. Indeed, possible crosstalk between translation and mRNA decay in *T. brucei* was noted previously (*Antwi et al., 2016*). We propose that optimal-codons reduce mRNA turnover in trypanosomes by increasing the rate of translation, increasing the association with actively translating ribosomes, and thereby protecting mRNA from degradation. In this scenario, slow translation or stalling would promote mRNA degradation (*Deana and Belasco, 2005*). Indeed, mRNA decay is known to be co-translational in yeast (*Hu et al., 2009*; *Pelechano et al., 2015*).

Mechanisms by which optimal codons impact ribosome translation rates remain a subject of intense debate (*Hanson and Coller, 2018*). Our findings in *T. brucei* suggest a more pronounced impact on translation that also impacts mRNA stability. For reference, estimates for highly expressed genes in yeast yield initiation rates of once every two seconds and elongation rates up to twenty amino acids per second (*Futcher et al., 1999*). In *T. brucei*, it is estimated that there are 125,000 ribosomes and 19,000 non-*VSG* mRNA molecules per bloodstream form cell (*Haanstra et al., 2008*). Notably, a number of studies point to translation initiation as the rate-limiting step in protein synthesis (*Jackson et al., 2010*) and ribosome occupancy was found to be similar on optimised and non-optimised constructs in yeast (*Presnyak et al., 2015*). In simple terms, a constant translation initiation rate combined with variable elongation rates would be expected to yield rapidly translated mRNAs loaded with relatively few ribosomes and slowly translated mRNAs loaded with many, potentially stalled, ribosomes. In contrast to this scenario, however, our analysis indicates correspondence between optimal codons and ribosome occupancy. This suggests a coupling between elongation and initiation in trypanosomes which could be explained by slow ribosomes affecting trailing ribosomes and decreasing the initiation rate (*Chu et al., 2014*; *Hanson and Coller, 2018*) or by ribosome recycling on the same mRNA [see Supplementary information S5 in (*Jackson et al., 2010*).

It should be possible in the future to further refine mathematical models of gene expression and also to better understand the relative impact and evolution of codon usage in trypanosomatids. Improvements in quantitative proteomics, measurements of protein synthesis and turnover rates, a better understanding of the impact of untranslated sequences on translation and mRNA turnover and an improved understanding of the relative contributions of individual codons to translation elongation and initiation will all help make this possible. We conclude that protein-coding sequence composition contributes to differences in steady-state mRNA and protein abundance in trypanosomatids. The ability to predict relative expression, based on protein-coding sequences alone, indicates that codon usage makes a major contribution to the transcriptomes and proteomes of these cells. Our results support a model whereby translation rate is increased by optimal codons, resulting in reduced mRNA turnover.

# Materials and methods

**Key resources table**

| Reagent type (species) or resource | Designation | Source or reference | Identifiers | Additional information |
|---|---|---|---|---|
| gene (*Gaussia princeps*) | *gLUC* | PMID: 18408930 | AY015993.1 | wild-type |
| gene (*Gaussia princeps*) | *gLUC* | PMID: 18408930 | EU372000 | human codon-optimised |
| cell line (*Trypanosoma brucei*) | 2T1 | PMID: 16182389 | | |
| transfected construct (*Trypanosoma brucei*) | pRPa-iSL plasmid | PMID: 18588918 | 69244 | available from addgene.org |
| transfected construct (*Trypanosoma brucei*) | pRPa plasmid | PMID: 18588918 | | |
| transfected construct (*Trypanosoma brucei*) | pRPa-λ plasmid | this paper | | see materials and methods |
| antibody | α-gLUC | New England Biolabs | | one in 1000 |
| sequence-based reagent | EUluc5 oligonucleotide | this paper | | GATCCTGCAGCTCGAGATGAAGCCC ACCGAGAACAACG |

*Continued on next page*

*Continued*

| Reagent type (species) or resource | Designation | Source or reference | Identifiers | Additional information |
|---|---|---|---|---|
| sequence-based reagent | EUluc3 oligonucleotide | this paper | | GATC*GAATTC*AGATC*TAAGCTT*TTA *CAGCTTCGA*GTCGCCGCCGGCGCC |
| sequence-based reagent | WTluc5 oligonucleotide | this paper | | GATC*CTCGAG*ATGAAACCA ACTGAAAACAATG |
| sequence-based reagent | WTluc3 oligonucleotide | this paper | | GATC*AAGCTT*TT*ATAATTTACT*ATCAC CACCGGCACCCTT |
| sequence-based reagent | Lambda5 oligonucleotide | this paper | | GATC*AAGCTT*GCAGGGT GAGATTGTGGC |
| sequence-based reagent | Lambda3 oligonucleotide | this paper | | GATC*GAATTC*GCTCAGTT GTTCAGGAATATG |
| sequence-based reagent | TUBF oligonucleotide | this paper | | AGATCTTCAAACACTAGTTTAAGC |
| sequence-based reagent | TUBR oligonucleotide | this paper | | CATGATAAATAAATAGA AGTGCTTTGTTG |
| sequence-based reagent | λF oligonucleotide | this paper | | GATTCATAAGTTCCGCT GTGTGCCGCATCTC |
| sequence-based reagent | λR oligonucleotide | this paper | | GCTCAGTTGTTCAGGAA TATGGTGCAGCAG |
| commercial assay or kit | BioLux Gaussia luciferase | New England Biolabs | | |
| software, algorithm | Bowtie 2 | PMID: 22388286 | | |
| software, algorithm | SAMtools | PMID: 19505943 | | |
| software, algorithm | edgeR | PMID: 19910308 | | |
| software, algorithm | CAI calculator | http://www.umbc.edu// codon/cai/cais.php | | |
| software, algorithm | ANACONDA | http://bioinformatics.ua.pt/ software/anaconda/ | | |
| online database | TriTrypDB, RRID:SCR_007043 | http://tritrypdb.org/ tritrypdb/ | | |

## *T. brucei* strains and plasmids and genetic manipulation

Bloodstream form *T. brucei*, Lister 427 (MITat 1.2), clone 221a cells and 2T1 cells (*Alsford et al., 2005*) have been confirmed mycoplasma free and their identity was confirmed by RNA-seq (see below). These cells were grown in HMI-11 medium and genetically manipulated using electroporation as described (*Alsford et al., 2005*). Initially, human^opt and WT *Gaussia princeps* gLUC genes were introduced into a *T. brucei* inducible expression vector. Human^opt gLUC (EU372000) was amplified using the EUluc5 and EUluc3 primers (Key Resources Table; *Pst*I/*Eco*RI restriction sites are underlined) and cloned between *T. brucei* *TUB* mRNA processing signal sequences. The entire cassette was then cloned (*Bam*HI/*Bsp*120I) in pRPa^iSL (*Alsford and Horn, 2008*). WT gLUC (AY015993.1) was amplified from pUC19GLuc (Prolume Ltd/NanoLight) using the WTluc5 and WTluc3 primers (Key Resources Table; *Xho*I/*Hin*dIII restriction sites are underlined) and used to replace the human^opt gLUC gene in the pRPa^iSL backbone. This also allowed us to remove an *N*-terminal signal-sequence and to add a *C*-terminal peroxisome-targeting signal (SKL) to both proteins (see *italics* in the Key Resources Table in EUluc3 and WTluc3). A 443 bp λ-DNA fragment was then amplified using the Lambda5 and Lambda3 primers (Key Resources Table; *Hin*dIII/*Eco*RI restriction sites are underlined) and cloned immediately downstream of the gLUC coding sequence. Once we had established that gLUC expression was not toxic, we moved the expression cassette (*Bam*HI/ *Bsp*120I) to pRPa (*Acc*65I/*Bsp*120I), a vector for constitutive expression (*Alsford and Horn, 2008*). New gLUC and GFP genes were designed using a previously published codon usage table (*Horn, 2008*). These genes were synthesised (Genscript) and cloned (*Xho*I/*Hin*dIII) in the constitutive

pRPa$^\lambda$ expression vector. All of these constructs were linearized with *AscI* prior to electroporation and insertion at the ribosomal DNA spacer locus on chromosome 2. Tetracycline was applied at 1 µg.ml$^{-1}$ for 24 hr to induce expression in pRPa$^i$-based strains.

## Protein analysis and luciferase activity assay

Total cell extracts equivalent to $1 \times 10^6$ cells were separated on SDS-polyacrylamide gels and subject to either standard or LICOR western blotting analysis according to the manufacturers' instructions. For standard western blots, duplicate gels were generated and one was stained with Coomassie and the other was used to produce the nitrocellulose blot. Blots were blocked in 5% milk in TBST and washes were performed in TBST (0.05% Tween). Blots were then probed with 1/1000 α-gLUC primary antibody (New England Biolabs) and 1/2000 α-rabbit secondary antibody (Bio-Rad). For LICOR blots, blocking was performed in 50 mM Tris, pH 7.4, 0.15 M NaCl, 0.25% BSA, 0.05% Tween, 2% fish skin gelatine (Sigma, UK) and washes were performed in TBST (α-gLUC) or PBST (α-GFP). Nitrocellulose blots were incubated with 1/1000 α-gLUC or 1/5000 α-GFP (Life Technologies) and 1/20 α-tubulin (kind gift from Keith Gull), followed by 1/10,000 α-mouse and 1/10,000 α-rabbit IR Dye antibodies (LICOR). Lysates for gLUC assays were prepared by adding 20 µl of $1 \times$ luciferase cell lysis buffer (New England Biolabs) to $2 \times 10^6$ pelleted cells and these were directly used to perform BioLux Gaussia luciferase assays (New England Biolabs) according to the manufacturers' instructions and using a TopCount plate-reader with white-walled plates. One-way ANOVA tests were carried out in GraphPad Prism (version 7).

## RNA analysis

For RNA extraction, $5 \times 10^7$ bloodstream form *T. brucei* were collected and RNA prepared using the Qaigen RNeasy kit, according to the manufacturer's instructions. The TUB and λ probes were generated by PCR, using the TUBF and TUBR and λF and λR primers, respectively (see Key Resources Table). RNA blotting was carried out according to standard protocols and signal quantification was carried out using a phosphorimager. One-way ANOVA tests were carried out in GraphPad Prism (version 7).

## RNA-seq

For RNA-seq, RNA samples were prepared from three independent bloodstream form *T. brucei* cultures; batches of $5 \times 10^7$ cells at a density of $1–2 \times 10^6$/ ml. RNA was prepared using the Qiagen RNeasy kit, according to the manufacturer's instructions. RNA samples were assessed using the TapeStation Platform and the QuBit platform and all samples were normalised for an input of 1 µg. Polyadenylated transcripts were enriched using oligo d(T)$_{25}$ magnetic beads (NEB). We used the NEBNext Ultra II Directional RNA Library Prep Kit for Illumina platforms which is optimised for ~200 bp inserts. Following PCR enrichment and purification each sample library was again QC checked using TapeStation and Qubit. 300 ng of each sample were pooled to generate a pooled library of 233 nM. The pooled library was diluted to 4 nM before preparation for running on the NextSeq platform. Samples were loaded at 1.6 pM on a $2 \times 75$ bp High Throughput Flowcell achieving a % >=Q30, 81.9G, 93.0%. Reads were mapped to the *T. brucei* 927 reference genome (TriTrypDB, RRID:SCR_007043). Bowtie 2-mapping (*Langmead and Salzberg, 2012*) was as previously described (*Glover et al., 2016*) with the parameters –very-sensitive –no-discordant –phred33 (no mismatches allowed). Approximately 115 million reads were aligned. Alignment files were manipulated using SAMtools (*Li et al., 2009*). Read counts were normalised using edgeR (*Robinson et al., 2010*).

## Codon analysis

The highly expressed *T. brucei* reference set comprised those genes (n = 50) encoding proteins of >250 amino acids that registered >25 unique peptides and >75% coverage in the proteome dataset (*Urbaniak et al., 2012*). We used an online CAI calculator (umbc.edu/codon/cai/cais.php) to generate CAI values. Codon pair bias was analysed using the ANACONDA (bioinformatics.ua.pt/software/anaconda/) software package (*Moura et al., 2011*). The *T. brucei* proteome analysis set comprised those non-redundant genes encoding proteins that registered >3 unique peptides in the proteome dataset (*Urbaniak et al., 2012*). The *T. brucei* transcriptome analysis set comprised all non-redundant genes; both sets excluded genes transcribed by RNA polymerase I. The highly

expressed *T. vivax* reference set comprised those genes encoding proteins that registered >20 unique peptides and >15 peptides/kbp in the proteome dataset; data merged from the three life cycle stages (*Jackson et al., 2015*). The *T. vivax* proteome analysis set comprised those non-redundant genes encoding proteins that registered >3 unique peptides in the proteome dataset and that were also orthologous to genes in the *T. brucei* set. The *T. vivax* transcriptome analysis set comprised genes that were orthologous to those in the *T. brucei* set and that also registered >10 tags. The highly expressed *L. mexicana* reference set comprised orthologues of the equivalent *T. brucei* set that also registered >100 FPKM; data merged from the three life cycle stages (*Fiebig et al., 2015*). The *L. mexicana* transcriptome analysis set comprised genes that were orthologous to those in the *T. brucei* set and that also registered >5 FPKM. Excel functions were used to derive Pearson correlation coefficients and trend-lines.

## Acknowledgements

We thank Ralph Bock (Max Planck Institute) for the human codon optimized *Gaussia* luciferase gene, Sebastian Hutchinson (Institut Pasteur) and Lucy Glover (Institut Pasteur) for discussions and assistance, Mike Ferguson (University of Dundee) and Mick Urbaniak (University of Lancaster) for advice on analysis of the proteome data, Keith Gull (University of Oxford) for the anti-tubulin antibody and Andrew Cassidy (University of Dundee) for assistance with RNA-seq. This work was supported by a Wellcome Senior Investigator Award (100320/Z/12/Z to DH). Our work is also supported by a Wellcome Centre Award (203134/Z/16/Z).

## Additional information

### Funding

| Funder | Grant reference number | Author |
| --- | --- | --- |
| Wellcome Trust | 100320/Z/12/Z | David Horn |
| Wellcome Trust | 203134/Z/16/Z | David Horn |

The funders had no role in study design, data collection and interpretation, or the decision to submit the work for publication.

### Author contributions

Laura Jeacock, Conceptualization, Formal analysis, Investigation, Methodology, Writing—original draft; Joana Faria, Data curation; David Horn, Conceptualization, Formal analysis, Supervision, Funding acquisition, Investigation, Methodology, Writing—original draft, Project administration, Writing—review and editing

### Author ORCIDs

David Horn http://orcid.org/0000-0001-5173-9284

### Decision letter and Author response

Decision letter https://doi.org/10.7554/eLife.32496.019
Author response https://doi.org/10.7554/eLife.32496.020

## Additional files

### Supplementary files

• Supplementary file 1. Sheet 1: Synthetic genes. Sequences of *gLUC* and *GFP* genes with high, medium or low proportions of GC3-codons. Sheet 2: *T. brucei* expression data. CAI values, proteome and transcriptome data and predicted expression levels are tabulated for the non-redundant gene sets analysed in *Figures 5–6*. Sheet 3: *T. brucei* gene cohorts. Data for the genes analysed in *Figure 7*. Sheet 4: *T. brucei* - extended set of ranked CAI values and predictions, including GeneID and product description; n = 8479. Sheet 5: *T. vivax* - ranked CAI values and predictions, including

GeneID and product description; n = 7836. Sheet 6: *L. mexicana* - ranked CAI values and predictions, including GeneID and product description; n = 5715.
DOI: https://doi.org/10.7554/eLife.32496.014

• Transparent reporting form
DOI: https://doi.org/10.7554/eLife.32496.015

## Major datasets
The following dataset was generated:

| Author(s) | Year | Dataset title | Dataset URL | Database, license, and accessibility information |
|---|---|---|---|---|
| Laura J, Joana F, David H | 2018 | T. brucei RNA-seq data | https://www.ebi.ac.uk/ena/data/view/PRJEB22797 | Publicly available at the European Nucleotide Archive (accession no. PRJEB22797) |

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
