## [Decision Letter]

Thank you for sending your article entitled "Codon usage bias controls mRNA and protein abundance in the African trypanosome" for peer review at *eLife*. Your article is being evaluated by three peer reviewers, and the evaluation has been overseen by a Reviewing Editor and Wendy Garrett as the Senior Editor.

Given the list of essential revisions, including new experiments, the editors and reviewers invite you to respond within the next two weeks with an action plan and timetable for the completion of the additional work. We plan to share your responses with the reviewers and then issue a binding recommendation.

Essential revisions:

1) RNASeq correlations:a) For all plots in which mRNA levels are compared with the codon usage, include *all* mRNA abundances, not just the subset for which protein measurements are available. There is no rationale for excluding mRNAs that give levels of protein that are not detectable by mass spectrometry. This step excludes a substantial proportion of the transcriptome.

b) Please plot the CAI against several datasets other than your own. If the correlation is only true for your own dataset, then it is no use.

c) The authors hypothesis is that certain codons "reduce mRNA turnover". To test this they must plot the CAI against the measured mRNA turnover rates (10.1111/mmi.12764). Of course there will be discrepancies, some of which will have been caused by the way the half-life information was obtained, and these can be discussed. Basically, the authors need to guide the reviewer through the published data and discuss how their data differs/ overlaps, etc. RNA preparation methods are known to have effects, and that information should be brought into the Discussion.

The interpretation must be toned down throughout and all over-statements removed. The paper shows *only* that codon usage "contributes to" mRNA and protein abundance. It does not show that it is "the major factor". To support such a strong conclusion, a careful calculation needs to show that codon usage contributes to variance more than every other possible factor.

2) Figure 7, showing different complexes, should be expanded to include other mutli-subunit complexes and additional metabolic pathways. In addition, the statement in the Abstract about 0.91 corr. Is very misleading and needs to be removed. It only shows the same relationship as was already seen, but for a small, cherry-picked subset of genes. Including a smaller number of genes will automatically improve the correlation coefficient – this is a statistical artifact.

Moreover, developmental co-regulation of mRNAs encoding different complexes was already shown using microarrays (Queiroz et al), and the distributions of mRNA half-lives for the complexes are already available (10.1111/mmi.12764), why is this not referred to?. For such complexes, protein abundance may well also be regulated by selective degradation of uncomplexed subunits.

3) Data analysis and significance:a) Relating to Figure 7, It is not clear how the cohorts were put together nor why certain cohorts were selected and others not? E.g. why was a cohort for RNA pol II shown but not for RNA pol I?

b) To strengthen the claim that the CAI can be used to predict mRNA and protein levels, the authors should determine how well it can be used to predict RNA levels in PF. There is no need to generate new RNAseq data, the analysis can be done using publicly available RNAseq datasets.

c) Methods:

The authors should follow the ENCODE guidelines on RNA-Seq reporting so that the library preparation can be reproduced:https://www.encodeproject.org/about/experiment-guidelines/

• For example, what polY-dT beads were used?

• Magnetic or cellulose? From what manufacturer?

• What kit was used for library preparation etc.?

• What percentage of reads could be aligned?

4) The Discussion and working model are not clear.a) The authors argue that if translation elongation is the rate-limiting step in protein synthesis, optimal codons leading to faster elongation will have a major impact on the rate of protein synthesis. While this argument is not incorrect, it neglects a vast amount of literature pointing to translation initiation as the rate-limiting step, not elongation (Jackson et al., 2010; Shah et al., Cell 153, 1589-1601 2013).

The authors should address the possibility that the primary mechanism by which codon usage affects protein synthesis is not via the rate of elongation but by modulating RNA stability.

b) The authors write:

"We propose that GC3-codons reduce mRNA turnover by increasing the rate of translation, increasing the association with actively translating ribosomes, and thereby protecting mRNA from attack by nucleases."

This hypothesis is contradicted by a statement in paragraph two of the Discussion where the authors write that genes encoding ribosomal proteins have a particularly high CAI score (which should lead to a high elongation rate), a reduced ribosome density and high stability. The literature in the codon usage field shows that increased rate of elongation does not lead to changes in polysome profiles (see Presnyak, 2015). Although this is somewhat counter intuitive, it is the finding in other eukaryotic systems. What should be mentioned are those data and the fact that the issue of translation initiation rates are never discussed in this pare, or the literature at large.

However, the hypothesis that slow translation or stalling would promote mRNA degradation is supported by data in other organisms and seems reasonable.

The authors should check the logic of their arguments so that either all experimental data is explained by one hypothesis or, if this is not possible, conflicting observations are discussed.

Review the literature on links between translation and mRNA decay in other organisms more comprehensively – some important references have been omitted.

[Editors' note: the authors’ plan for revisions was approved and the authors made a formal revised submission.]

---

## [Author Response]

Essential revisions:1) RNASeq correlations:a) For all plots in which mRNA levels are compared with the codon usage, include all mRNA abundances, not just the subset for which protein measurements are available. There is no rationale for excluding mRNAs that give levels of protein that are not detectable by mass spectrometry. This step excludes a substantial proportion of the transcriptome.

mRNA plots, Figure 5 and Figure 6, now include all mRNA abundances of >1 RPKM (from n=2348 to n=7225).

b) Please plot the CAI against several datasets other than your own. If the correlation is only true for your own dataset, then it is no use.

(Length-adjusted) CAI is now plotted against data from:

Hutchinson et al., 2016: Figure 6—figure supplement 1.

Christiano et al., 2017: Figure 6—figure supplement 1.

Vasquez et al., 2014: Figure 8.

Fadda et al., 2014: Figure 8.

Jackson et al., 2015: Figure 9.

Fiebig et al., 2015: Figure 9.

We see a correlation in every case and have therefore changed ‘African trypanosomes’ to ‘trypanosomatids’ in the title. To support these new data, there is a new sentence in the Abstract beginning ‘Our estimates also […]’; two new sub-sections at the end of the Results section, new text in the Discussion (details below), at the end of the Materials and methods and new datasets in Supplementary file 1; sheets five and six.

c) The authors hypothesis is that certain codons "reduce mRNA turnover". To test this they must plot the CAI against the measured mRNA turnover rates (10.1111/mmi.12764). Of course there will be discrepancies, some of which will have been caused by the way the half-life information was obtained, and these can be discussed. Basically, the authors need to guide the reviewer through the published data and discuss how their data differs/ overlaps, etc. RNA preparation methods are known to have effects, and that information should be brought into the Discussion.

See response 1b above: Analysis of data from Fadda et al., 2014 is now included in Figure 8.

The interpretation must be toned down throughout and all over-statements removed. The paper shows only that codon usage "contributes to" mRNA and protein abundance. It does not show that it is "the major factor". To support such a strong conclusion, a careful calculation needs to show that codon usage contributes to variance more than every other possible factor.

These statements have been adjusted as suggested.

2) Figure 7, showing different complexes, should be expanded to include other mutli-subunit complexes and additional metabolic pathways. In addition, the statement in the Abstract about 0.91 corr. Is very misleading and needs to be removed. It only shows the same relationship as was already seen, but for a small, cherry-picked subset of genes. Including a smaller number of genes will automatically improve the correlation coefficient – this is a statistical artifact.Moreover, developmental co-regulation of mRNAs encoding different complexes was already shown using microarrays (Queiroz et al), and the distributions of mRNA half-lives for the complexes are already available (10.1111/mmi.12764), why is this not referred to? For such complexes, protein abundance may well also be regulated by selective degradation of uncomplexed subunits.

Figure 7 has been expanded (also see response 3 below). We focus mainly on constitutive complexes in this figure rather than the developmentally regulated complexes described in Queiroz et al., 2009. We’ve now also expanded our analysis based on this figure and discuss key complexes in more detail; see paragraph beginning ‘Using our relative […]’ (Results, section five). “0.91” has been removed. We previously mentioned possible “destabilisation of unassembled subunits” (Discussion, now end of third paragraph).

3) Data analysis and significance:a) Relating to Figure 7, It is not clear how the cohorts were put together nor why certain cohorts were selected and others not? E.g. why was a cohort for RNA pol II shown but not for RNA pol I?

Our analysis in Figure 7 was designed to address the under-sampling problem and we; therefore, only analyzed complexes with eight or more components (RNA Pol-I comprises seven known components), also excluding cohorts subject to cell cycle restricted expression or major developmental regulation. We have now added fifteen additional complexes (including RNA Pol-I) and also now include PMIDs for each complex in Supplementary file 1.

b) To strengthen the claim that the CAI can be used to predict mRNA and protein levels, the authors should determine how well it can be used to predict RNA levels in PF. There is no need to generate new RNAseq data, the analysis can be done using publicly available RNAseq datasets.

See response 1b above: Analyses of data from Hutchinson et al., 2016 and Christiano et al., 2017 are now included.

c) Methods:The authors should follow the ENCODE guidelines on RNA-Seq reporting so that the library preparation can be reproduced:https://www.encodeproject.org/about/experiment-guidelines/• For example, what polY-dT beads were used?• Magnetic or cellulose? From what manufacturer?• What kit was used for library preparation etc.?• What percentage of reads could be aligned?

We have now included a more detailed RNA-seq protocol (Materials and methods, RNA-seq section). Figure 5—figure supplement 1 indicates percentage of reads aligned.

4) The Discussion and working model are not clear.a) The authors argue that if translation elongation is the rate-limiting step in protein synthesis, optimal codons leading to faster elongation will have a major impact on the rate of protein synthesis. While this argument is not incorrect, it neglects a vast amount of literature pointing to translation initiation as the rate-limiting step, not elongation (Jackson et al., 2010; Shah et al., Cell 153, 1589-1601 2013).The authors should address the possibility that the primary mechanism by which codon usage affects protein synthesis is not via the rate of elongation but by modulating RNA stability.

We have read the suggested review and theory articles and now discuss the relationship between initiation and elongation in more detail, also citing Jackson et al., 2010 (Discussion, paragraph 6). This text also addresses the elongation v. stability point (also see Results, end of section 2; sentence beginning ‘While the range of values […]’).

b) The authors write:"We propose that GC3-codons reduce mRNA turnover by increasing the rate of translation, increasing the association with actively translating ribosomes, and thereby protecting mRNA from attack by nucleases."This hypothesis is contradicted by a statement in the second paragraph of the Discussion where the authors write that genes encoding ribosomal proteins have a particularly high CAI score (which should lead to a high elongation rate), a reduced ribosome density and high stability. The literature in the codon usage field shows that increased rate of elongation does not lead to changes in polysome profiles (see Presnyak, 2015). Although this is somewhat counter intuitive, it is the finding in other eukaryotic systems. What should be mentioned are those data and the fact that the issue of translation initiation rates are never discussed in this pare, or the literature at large.However, the hypothesis that slow translation or stalling would promote mRNA degradation is supported by data in other organisms and seems reasonable.The authors should check the logic of their arguments so that either all experimental data is explained by one hypothesis or, if this is not possible, conflicting observations are discussed.Review the literature on links between translation and mRNA decay in other organisms more comprehensively – some important references have been omitted.

A slightly modified version of the first statement now appears at the end of paragraph five in the Discussion. The section on ribosome profiling has been revised and now appears in paragraphs five and six in the Discussion; section beginning ‘Ribosome profiling, nascent chain profiling […]’, while a statement on ribosomal protein transcripts now appears in the Results (section 4; sentence beginning ‘This may reflect […]’. Indeed, the behavior of these particular transcripts is unusual as reported previously by Antwi et al. 2016. The point relating to ribosome profiles is particularly interesting. We analyzed ribosome profiling data from Vasquez et al., 2014 and see correspondence between translation efficiency and our ‘expression level’ predictions (Figure 8). This is now discussed in relation to the findings of Presnyak et al., 2015; using synthetic genes in yeast (Discussion, paragraph six; sentence beginning ‘Notably, a number of […]’). The other points regarding translation initiation and mRNA decay are addressed above.